# POViT: Vision Transformer for Multi-Objective Design and Characterization of Photonic Crystal Nanocavities

**DOI:** 10.3390/nano12244401

**Published:** 2022-12-09

**Authors:** Xinyu Chen, Renjie Li, Yueyao Yu, Yuanwen Shen, Wenye Li, Yin Zhang, Zhaoyu Zhang

**Affiliations:** 1Shenzhen Key Laboratory of Semiconductor Lasers, School of Science and Engineering, The Chinese University of Hong Kong, Shenzhen, 2001 Longxiang Ave, Shenzhen 518172, China; 2Shenzhen Research Institute of Big Data (SRIBD), 2001 Longxiang Ave, Shenzhen 518172, China; 3School of Data Science, The Chinese University of Hong Kong, Shenzhen, 2001 Longxiang Ave, Shenzhen 518172, China

**Keywords:** deep learning, vision transformer, nanophotonics, lasers, photonic crystals

## Abstract

We study a new technique for solving the fundamental challenge in nanophotonic design: fast and accurate characterization of nanoscale photonic devices with minimal human intervention. Much like the fusion between Artificial Intelligence and Electronic Design Automation (EDA), many efforts have been made to apply deep neural networks (DNN) such as convolutional neural networks to prototype and characterize next-gen optoelectronic devices commonly found in Photonic Integrated Circuits. However, state-of-the-art DNN models are still far from being directly applicable in the real world: e.g., DNN-produced correlation coefficients between target and predicted physical quantities are about 80%, which is much lower than what it takes to generate reliable and reproducible nanophotonic designs. Recently, attention-based transformer models have attracted extensive interests and been widely used in Computer Vision and Natural Language Processing. In this work, we for the first time propose a Transformer model (POViT) to efficiently design and simulate photonic crystal nanocavities with multiple objectives under consideration. Unlike the standard Vision Transformer, our model takes photonic crystals as input data and changes the activation layer from GELU to an absolute-value function. Extensive experiments show that POViT significantly improves results reported by previous models: correlation coefficients are increased by over 12% (i.e., to 92.0%) and prediction errors are reduced by an order of magnitude, among several key metric improvements. Our work has the potential to drive the expansion of EDA to fully automated photonic design (i.e., PDA). The complete dataset and code will be released to promote research in the interdisciplinary field of materials science/physics and computer science.

## 1. Introduction

Recently, Deep Learning (DL) has been widely utilized in multiple fields such as medical imaging [1,2], natural language processing (NLP) [3,4], autonomous driving [5], face recognition [6] and object detection [7]. Deep neural networks (DNN) can process data effectively and learn rich representations by its strong capacity in dealing with high-dimensional massive data. Impressed by the application prospects of DL, some very recent works have tried Multi-Layer Perceptrons (MLPs) [8,9,10,11] and Convolutional Neural Networks (CNNs) [8,12,13,14,15,16,17,18] to design and optimize optoelectronic devices such as nanoscale lasers.

The behavior of nanoscale lasers can be characterized by calculating the material gain in the quantum well/dot and the transverse/longitudinal modes in the defect microcavity [19,20]. However, the traditional method for designing nanoscale lasers is usually time-costing and inefficient because all the physical parameters are adjusted manually via simulation tools such as COMSOL and Lumerical, whose finite-difference time-domain (FDTD) or finite element analysis (FEA) method is computationally intensive. Moreover, gradient-based optimization methods will often face difficulties in convergence because of the high-dimensional parameter space associated with physical systems and the presence of multiple local minima [10]. Such complicated design depends heavily on computational tractability and designers’ extensive experience [21]. Thus, if DL can be successfully applied to this field, there is no doubt that it will save tremendous amount of effort and resources in designing a well-formulated photonic device.

However, it seems that traditional DL models such as CNNs and MLPs (Figure 1b) are confronted with their performance bottleneck when tasked with designing highly complex physical systems. For example, it is still quite hard to increase the correlation coefficient of prediction results by adjusting DNN’s hyperparameters or adopting gradient-based optimization algorithms only. That is why the Vision Transformer, which is a cutting-edge technique based on a unique attention mechanism [3], has emerged as a disruptive alternative in deep learning. Empowered by the transformer’s outstanding performance in a variety of engineering applications, to the best of our knowledge, this paper is the first to investigate existing self-attention models including the original Vision Transformer (ViT) [22], Convolutional Vision Transformer (CvT) [23], and our own version of ViT applied to designing and characterizing Photonic Crystal (PC) nanocavities. PCs are core components of high-performance nanoscale semiconductor lasers used in next-gen Photonic Integrated Circuits (PIC) and LiDAR [20,24,25,26,27,28]. We thereafter name our final deep learning model *POViT*: *P*h*O*tonics *Vi*sion *T*ransformer.

### 1.1. Contributions

This paper studies a new technique for solving the fundamental challenge in nanophotonic design: fast and accurate characterization of nanoscale photonic devices with minimal human intervention. In particular, we for the first time propose a transformer-based DL model (POViT) to efficiently design and simulate photonic crystal nanocavities with multiple objectives under consideration. In this paper, multi-objective means the ability to predict more than one photonic/electromagnetic property of the PC nanocavity being characterized, which in our case are the quality factor *Q* and modal volume *V*. In the fabrication process of photonic crystals, POViT may become a promising alternative to existing time-consuming simulation tools such as FDTD and FEA and has the potential to replace trial-and-error or manual design approaches by humans. In short, the speed and accuracy of POViT demonstrated here can offer us a possibility to streamline the design process of PCs unachievable with conventional methods, while at the same time yield high-quality PC designs suitable for nanolasers.

We demonstrate the strength of the proposed POViT by comparing it against the performance of SOTA CvT, CNN and MLP models at predicting the quality factor *Q* and modal volume *V* of PC nanocavities. We found that our POViT successfully beats the above models where the test losses decrease largely (to $Q_{mse\;loss}=0.000116$ & $V_{mse\;loss}=0.001114$) and correlation coefficients improved dramatically (to $V_{coeff}=92.0\%$) with the minimum prediction error still remaining minuscule ($Q_{pred\;err}=0.000035\%$ & $V_{pred\;err}=0.000961\%$). A full list of metric improvements reported by the presented POViT model is tabulated and articulated later in this text. An overview of the history of DL models is graphically illustrated in Figure 1b as a timeline of the evolution of DNNs. In Figure 1b, DNNs have evolved from basic MLPs in the 1980s to the latest ViT based on attention mechanism circa. 2021.

Furthermore, we also conducted several experiments to prove the robustness of a special kind of activation layer, which is called absolute-value function (ABS) [29], that outputs the absolute values from the input. We have shown that the ABS layer can significantly improve the performances of ViT relative to conventional activation layers such as GELU [30]. Please see section III B in the Results & Discussion section for a detailed discussion on why ABS outperforms GELU.

In addition, we visualize the self-attention mechanism in the transformer blocks through heatmaps. The heatmaps indicate the contribution from different parts of the nanocavity structure to the laser device’s overall quality where regions in lighter colors need more attention from the POViT model.

Last but not least, our work paves the way for applying ViT to the rapid multi-objective characterization and optimization of nanophotonic devices without the need for major human intervention or trial-and-error iterations. Our methodology is inspired by the famous marriage of AI and Electronic Design Automation (EDA), a field extensively investigated by the academia and the industry alike. We mainly aim to empower the rise of fully automated photonic design through our efforts because the current state of Photonic Design Automation (PDA) is still largely lacking.

### 1.2. Related Work

The InAs/GaAs quantum dot PC nanocavity laser can be experimentally grown on a Silicon wafer substrate [26], but how to efficiently compute the quality (*Q*) factor of such a nanophotonic device is still an unsolved problem due to the high complexity of its physical structure. At the same time, it takes much time for FDTD-based simulation tools to simulate and evaluate the optical properties of the targeted structure. Recently, CNNs [18] have been proposed to train and predict the *Q* factor with a small training dataset (about 1000 samples) and the model did not consider the impact of air hole radius on *Q*. The prediction error is about 16%, which is not reliable to be utilized in real practice. Built upon recent progress made by Asano et al. [18], some works [8] reported that the performance of CNN models could be improved by a larger dataset. Besides the *Q* factor, modal volume *V* is also an important parameter for evaluating a nanolaser’s performance and attributes and is crucial for reducing the device footprint and achieving tight on-chip integration [18]. However, predicting *V* was not accomplished in the above cited literature [8,13,18].

To take the modal volume *V* into consideration, authors in recent years [14] succeeded in training and predicting *Q* and *V* simultaneously and maintained small test losses, which makes it the state-of-the-art result at present. However, the correlation coefficient of *V* is still relatively low ($V_{coeff}=80.5\%$ in the test set [14]). The higher the coefficients $Q_{coeff}$ and $V_{coeff}$ are, the more accurate the model’s prediction results will be. Ideally, the best case should be where the coefficients are equal to 1 for the most reliable and reproducible design output. Hence, there is still a large gap left for improving $V_{coeff}$ by adopting better and more advanced DL models and algorithms.

Transformer models have demonstrated their power in various tasks, from NLP, computer vision (CV) to fundamental science areas. The Transformer was first introduced in NLP around 2017 [3,31] and later developed in CV in 2021 with allegedly better performance than CNN [22]. Many subsequent works attempted to modify the architecture of ViT [32,33,34] for better performance or apply transformer models to multidisciplinary research [7,35,36,37,38,39,40,41,42,43]. For example, a gated axial-attention model [38] was proposed to overcome the problem of lacking data samples in medical image segmentation. It extends the existing transformer architecture by adding control mechanism into the self-attention module. A Dual Attention Matching (DAM) module [43] was proposed to cover a longer video duration for enhanced event information learning and extraction, while the local temporal encodings are retrieved by the global Cross-Check mechanism. With temporal encodings between audio and visual signals co-existing, DAM can be conveniently applied to different audio-visual event localization problems. Detection transformer (DETR) [7] is an object detection model that solves a direct set prediction problem, which reasons about the relations of the objects and the global image context to directly output the final set of predictions in parallel. DETR was shown to significantly outperform competitive object detection baselines. A BERT-based multilingual model in bio-informatics treats DNA sequences as natural sentences and successfully identifies DNA enhancers [41]. Furthermore, a modified transformer network is applied to learn the semantic relationship between objects in collider physics [42].

Different from the original version of ViT [22], the Early CNN-embedded Vision Transformer (EarlyVT) replaces the linear layer before the transformer block with a convolutional embedding layer to split the input image into patches [32]. Another model, the Convolutional Vision Transformer (CvT) [23], not only uses the convolutional embedding layer but also substitutes linear projection layers in the Transformer block for depth-wise separable convolution operations.

## 2. Methods

### 2.1. Physical Structure of PC Nanocavity Laser

Our nanoscale laser is realized by the PC nanocavity shown in Figure 1a, which has a regular array of holes in a multi-layer semiconductor (i.e., Si and InP) slab. This particular structure is ultra powerful and efficient because the spontaneous lasing emission is substantially enhanced by manipulating electromagnetic wave propagation enabled by a photonic band gap [44], where photons will be gathered to form a laser beam because of the array of periodic air holes. These holes have a periodically different effective refractive index compared with the Indium Phosphide (InP) base, which makes photons easily captured and confined. Since the peripheral air holes are far away from the center, they will contribute little to the quality factor *Q* and modal volume *V*, i.e., these holes do not make a distinct change to the electromagnetic field when they are adjusted. For a quick overview of some of the actual semiconductor nanolasers fabricated by our group, refer to Appendix A in the Appendix A.

Out of simplicity and resource-friendly purpose, the modeling area only contains 54 holes, which are rounded by the white rectangle (see Figure 1a). For holes outside this rectangle, we keep them fixed to lower the computational cost. The lattice constant a=320 nm and the radius of air hole r=89.6 nm are the standard values, i.e., before changing air holes’ positions and radii, the distance between the center of every pair of adjacent holes is 320 nm, and the default hole radius equals 89.6 nm. The refractive index of InP slab is n=3.4, which may differ from other semiconductor materials.

### 2.2. Data Collection and Pre-Processing

The dataset is obtained from Li et al. [14] with 12500 samples (obtained under Apache License 2.0.) Each sample contains variations of positions and radii from 54 air holes in the PC structure as the input and the corresponding simulated results *Q* and *V* as the target. Before forwarding the data samples into the model, we reshape its size into N×3×5×12 where *N* refers to the batch size, “3” represents three channels (dx,dy,dr) of the holes and the numbers “5” and “12” denote the height and width of our PC (i.e., the array), respectively. This transformation will make the samples resemble actual images. Further details of the dataset are briefly dissected below to avoid any ambiguities:

Denote the original position of a hole as $\left(x_{0},y_{0}\right)$ and initial radius as $r_{0}$. We then randomly shift the positions horizontally and vertically together with the radius under a Gaussian distribution so that its position becomes $\left(x^{\prime },y^{\prime }\right)$ and radius is $r^{\prime }$. Define $dx=x^{\prime }-x_{0}$, $dy=y^{\prime }-y_{0}$, and $dr=r^{\prime }-r_{0}$. The Gaussian distribution of dx, dy, and dr as the input elements follows as:(1)$$\begin{array}{cc} & dx∼N(\mu =-8.7270\times 10^{-13},\sigma^{2}=5\times 10^{-10}) \end{array}$$(2)$$\begin{array}{cc} & dy∼N(\mu =\;\;\;3.3969\times 10^{-13},\sigma^{2}=5\times 10^{-10}) \end{array}$$(3)$$\begin{array}{cc} & dr∼N(\mu =-1.6978\times 10^{-12},\sigma^{2}=5\times 10^{-10}) \end{array}$$

Due to different numbers of holes in different rows in our PC (shown in Figure 1a), four extraneous zeros are added into the central row, together with two zeros at the top left & bottom right corners, to alight the input tensor’s column dimension. In practice, the training dataset size is 10,000 so that the remaining 2500 samples can be used as test data. i.e., the size of the test dataset is 2500. Furthermore, the 12,500 data samples are split randomly so that the features of data samples can be as diverse as possible, by which the generalization capabilities of the designed POViT can be maximized [8,45]. The sample distribution of the dataset is graphically illustrated in Appendix A in the Appendix A.

### 2.3. Architecture of POViT & CvT

The self-attention mechanism is a crucial part in the transformer. The input is projected into queries Q, keys K and values V by some linear projections. Transformer will search for the extant key-value pairs and add up these pairs by weights to give out the predictions. The scaled dot-product function of the attention layer is given as:(4)$$\begin{array}{ccc} A(Q,K,V) & = & \frac{QK^{T}}{\sqrt{d}}V \end{array}$$

The architecture of POViT is given in Figure 2. Since input size should be divisible by the patch size before patch embedding, which is chosen as 2, the input is added one row of zeros resulting in the tensor height increasing from 5 to 6. After that, input tensor will be sliced into several patches and processed by patch embedding and positional encoding sequentially to be transferred as token sequences into the transformer encoder.

Meanwhile, this paper compares two different activation functions–ABS and GELU (default)–in Feed Forward Layers (FFN) embedded in the transformer sublayers to examine which one has a better performance. Their expressions are listed below:(5)$$\begin{array}{ccc} \mathrm{GELU} & = & 0.5x(1+\tanh\left(\sqrt{\frac{2}{\pi }}\left(x+0.044715x^{3}\right)\right) \end{array}$$(6)$$\begin{array}{ccc} \mathrm{ABS} & = & \left|x\right| \end{array}$$

For the architecture of CvT, it bears a resemblance to the ViT except that the usual patch embedding layer that directly slices the image input into several pieces is replaced by a convolutional layer and the linear projections in the transformer block are adjusted to deep-wise separable convolution operations as well.

### 2.4. Model Performance Evaluation

To measure the performance of the models, the MSE losses (Equation (Equation 7)), minimum and converged prediction errors (Equation (Equation 8)), and correlation coefficients (Equation (Equation 9)) are calculated during the training process. The minimum prediction error is measured and recorded by our program at the test stage while the converged one will be averaged at the last few epochs. Consider the targets (label) of the dataset denoted as $t_{i}$ and corresponding prediction outputs marked as $p_{i}$. The model can be evaluated by:(7)$$\begin{array}{ccc} \mathrm{MSE} & = & \frac{1}{N}\sum_{i=1}^{N}{\left(t_{i}-p_{i}\right)}^{2} \end{array}$$(8)$$\begin{array}{ccc} \varepsilon_{pred} & = & \frac{\left|p_{i}-t_{i}\right|}{t_{i}}\times 100\% \end{array}$$(9)$$\begin{array}{ccc} \rho (t,p) & = & \frac{Cov(t,p)}{\sigma_{t}\sigma_{p}}=\frac{E[\left(t_{i}-\bar{t}\right)\left(p_{i}-\bar{p}\right)]}{\sqrt{\sum_{i=1}^{n}{\left(t_{i}-\bar{t}\right)}^{2}}\sqrt{\sum_{i=1}^{n}{\left(p_{i}-\bar{p}\right)}^{2}}} \end{array}$$

The Pearson correlation coefficient ρ(t,p)∈[−1,1] in Equation (Equation 9) can be utilized to measure the linearity between prediction results and targets. If the coefficient is close to 1, then the output will positively correspond to the target, which means the proposed model perfectly fits in this regression mapping problem.

## 3. Results & Discussion

### 3.1. Results

The purpose of the proposed POViT is to construct a reliable and efficient method to streamline the multi-objective design of nanophotonic devices. Initially, 10,000 data samples are chosen and shuffled randomly from the dataset and fed into the model, which runs for 300 epochs each time. After many rounds of experiments, the hyperparameters giving rise to the best performance are listed below. The initial learning rate lr=0.01, and the optimizer is *Adam* with the learning rate scheduler as MultiStepLR (milestone = [100, 160, 200] and gamma = 0.1). A comprehensive list of hyperparameters for POViT we used are reserved in Appendix A in the Appendix A. Results for the trained POViT using ABS and GELU, respectively, are illustrated in Figure 3 and Figure 4. The correlation coefficients of *Q* at both training and test appear to be the same in Figure 3 and Figure 4 because we kept only three significant figures after the decimal point. It also shows there is no overfitting during training the *Q* factor. Since correlation coefficients were not provided in these work [8,14] for CNNs, we procured the open-source code from the cited repo [8] (Code obtained under Apache License 2.0.), and expanded them to include the capability of predicting correlation coefficients. We found that test coefficients in CNNs are calculated to be $Q_{coeff}=98.7\%$ and $V_{coeff}=80.5\%$ (Appendix A in the Appendix A), respectively. From Figure 4 we can see the best test coefficients in the POViT model are $Q_{coeff}=99.4\%$ and $V_{coeff}=92.0\%$, which is 11.5% higher than the best result in previous CNN models.

To compare the performances of different models (CNN, MLP, POViT (ours), CvT) on the photonic characterization task, open-source code for CNN are downloaded from Li et al. [8] and augmented by us as stated above while results for MLP are followed directly from Li et al. [8] without modifications. Reproduced results for CNN are plotted in Appendix A in the Appendix A and original results for CvT in Appendix A in the Appendix A. Results of POViT and CvT are produced from scratch by us for this work. The dataset is the same with 12,500 samples in all during the experiments (see Appendix A in the Appendix A). Results for *Q* and *V* across different models are summarized and compared in Table 1. Furthermore, without harming the high test correlation coefficient $Q_{coeff}$, $V_{coeff}$ also dramatically increases where $V_{coeff}$ of POViT was improved to 92.0% (see Figure 4h) and $V_{coeff}$ of CvT is 88.8%. It indicates the proposed POViT can detect the relationship between L3 PC nanocavities’ structure and corresponding optical qualities precisely.

The advantages of the proposed POViT are exuded in the prediction accuracy, convergence speed, and linearity of the model’s correlation coefficients (see Figure 3 and Figure 4). The introduction of the self-attention mechanism was recently shown to surpass conventional CNNs, which used to be the SOTA in the computer vision arena. Furthermore, the convergence speed of POViT is fast because the MSE losses decrease to a low level in just 100 epochs and then remain at a stable state after that. Linearity of POViT, including CvT which combines the transformer as well, implies our model’s good robustness against noise disturbance.

To make experimental results with POViT more fair and reliable, each time the learning rate was changed, three trials were performed and the mean values with uncertainties are summarized (see Table 1). For POViT, the average value of $Q_{coeff}$ is above 99.0% and $V_{coeff}$ is around 90.0% with a small margin of error, which are notably improved relative to the other models. Furthermore, the improvements in prediction error (both min & conv) of POViT are substantially compared with previous CNN and MLP models where the minimum prediction error has been reduced by an order of magnitude, and converged error decreases by over 50%.

The relationships between learning rate (lr) and MSE loss, correlation coefficient, and prediction error of the proposed POViT embedded with ABS activation are analyzed and illustrated in Figure 5a–c and comparison between ABS and GELU represented by $V_{coeff}$ is shown in Figure 5d. For each plot, eight different learning rates are chosen and experiment for each learning rate is repeated three times to avoid outliers. Mean values from those three runs are calculated and plotted in Figure 5. In Figure 5c,d, when the learning rate is around 0.01, the average $V_{coeff}$ reaches a peak of 91.2%. Another sub-peak appears at lr=0.0002 where the average $V_{coeff}=90.5\%$. If the learning rate is larger than 0.01, performance of POViT will plunge greatly so only one more trial was performed at lr=0.02. We see that in Figure 5c,d, when 0.0002<lr<0.01, there is a valley of $V_{coeff}$ values which indicates the model has unluckily been trained to reach the local minimum. In consequence, we avoided using those “bad” $lr^{\prime }s$ in our final model. In Figure 5a, there is a narrow band of lr between 0.0005 and 0.001 where both *Q* and *V* reach minimum prediction errors. In Figure 5b, the average $V_{loss}$ fluctuates with tiny swings between lr=0.001 and 0.002, and it reaches a global minimum at lr=0.01.

As for the activation layer, which is embedded in the Feed Forward Network (FFN) in the transformer blocks (see Figure 2), our experiments found the absolute-value function (ABS) has a recognizably better performance than GELU when lr is relatively small (lr<0.0005). In Figure 5d, $V_{coeff}$ are plotted against lr to demonstrate a performance contrast in favor of the promising ABS activation layer. When the learning rate gets relatively large (lr≈0.001), there still exists a small gap where ABS retains an edge over GELU. After lr≥0.005 however, the curves of ABS and GELU almost overlap with each other, despite the fact that the curve of the former is always slightly above the latter. Based on the above observation, we conclude that ABS is superior to GELU for our applications.

Lastly, to explore which parts of the PC nanocavity tend to demand more attention from the six-layer POViT in predicting *Q* and *V*, we took a look at attenion heatmaps. Heatmaps are extracted via a visualizer of POViT during the training and test experiments (see Figure 6). With depth deepening (i.e., going from top to bottom), there exist more vertical line patterns on the attention maps, which means important information is aggregated to some specific tokens in our data. Within each layer (i.e., in the horizontal direction), where six MLP heads are chosen, we can observe some irregularities in the line patterns in those heatmaps. This indicates all the heads work fairly in coordination to produce the final predicted values.

### 3.2. Discussion

It is worthwhile to further study why activation function ABS outperforms GELU when lr is relatively small. Here, we provide a possible explanation—the dying ReLU phenomenon[46] or collapse to constant (C2C) [47]. Here, we present a somewhat detailed mathematical derivation to prove ABS’s advantages and why GELU suffers from C2C. The authors[47] introduce the *C* matrix as a characterization of C2C. First, we define the *C*-matrix,
(10)$$C_{L}\equiv C_{L}\left(x,\bar{x},\bar{W},\bar{b}\right):=\prod_{k=1}^{L}W_{k}^{T}D_{\phi }\left(z_{k},{\bar{z}}_{k}\right),$$
where $D_{\phi }\left(\cdot ,\cdot \right)\in R^{d\times d}$ is diagonal defined by,
(11)$${\left[D_{\phi }\left(u,v\right)\right]}_{ii}=\frac{{\left[\phi \left(u\right)-\phi \left(v\right)\right]}_{i}}{{\left[u-v\right]}_{i}},\;\;i=1,\cdots ,d,$$
and $\left\{z_{k}\right\}$ and $\left\{{\bar{z}}_{k}\right\}$ are computed via the recursion:(12)$$s_{0}=x;\;\;z_{k}=W_{k}s_{k-1}+b_{k},s_{k}=\phi \left(z_{k}\right),\;\;k=1,\cdots ,L,$$
and then, $F_{L}\left(x,W,b\right):=s_{L}$ defines a neural network function. Proposition 1 shows that C-matrices characterize the C2C phenomenon.

**Proposition** **1**([47])**.**
*Let network $F_{L}\left(x,\bar{W},\bar{b}\right):R^{d}\to R^{d}$ be defined as in (Equation 12). For any two distinct points $x,\bar{x}\in R^{d}$, there holds*
(13)$$F_{L}\left(x\right)-F_{L}\left(\bar{x}\right)={\left[C_{L}\left(x,\bar{x}\right)\right]}^{T}\left(x-\bar{x}\right).$$
*Consequently,*
(14)$$\lim_{L\to \infty }C_{L}\left(x,\bar{x}\right)=0\;\;⟹\;\;\lim_{L\to \infty }\left(F_{L}\left(x\right)-F_{L}\left(\bar{x}\right)\right)=0.$$

In Proposition 2 below, the authors [47] report that the probability for the absolute-value function to preserve distances in $R^{d}$ is much larger than that for ReLU.

**Proposition** **2**([47])**.**
*Suppose that $u,v\in R^{d}$ be i.i.d. random variables with*
$$\mathrm{Prob}\left({\left[u\right]}_{i}\geq 0\right)=\mathrm{Prob}\left({\left[v\right]}_{i}\geq 0\right)=p\in \left(0,1\right),\;\;i=1,\cdots ,d.$$
*Then*

(15)
$$\mathrm{Prob}\left(|\phi (u)-\phi (v)|=|u-v|\right)=\left\{\begin{array}{cc} p^{2d}, & \phi (t)=\max(0,t)\\ {\left(p^{2}+{\left(1-p\right)}^{2}\right)}^{d}, & \phi (t)=|t| \end{array}\right.$$



Considering the data distribution of dx, dy, and dr mentioned in section II B, input data gather in a small range after normalization and thus a considerable part of data elements are located on the negative half of the axis. As a result, ReLU-like activation functions (e.g., GELU) may suffer from C2C and some neurons in POViT could become inactive with weights reduced close to zero, which will be disadvantageous to the loss result. On the contrary, ABS, based on the above propositions, is less affected by C2C.

Next, fabrication error is also an important factor to consider in designing PCs, especially when dealing with nanometer-sized features. In order to make our design tolerant to fab errors, the retrieved *Q* factor and modal volume *V* should only loosely correlate with variations (say, on the order of tens of nm) in design parameters; in other words, shifting the design parameters should not lead to drastic changes in *Q* and *V*. This tolerance robustness of POViT will be studied in future works by us.

Lastly, although the proposed POViT has the edge over other models at its fast and precise multi-objective design and characterization, it still has room for improvement, especially in increasing the correlation coefficient $V_{coeff}$ and converged prediction error of *V*. Future works can be put on fine-tuning the model’s hyperparameters, such as the attention layers’ depth or do trials on other optimizers and learning rate schedulers.

## 4. Conclusions

The proposed POViT is the first to introduce Vision Transformer into designing and characterizing nanophotonic devices to the best of our knowledge. Based on a self-attention mechanism, POViT successfully predicts multiple objectives such as *Q* factor and modal volume *V* simultaneously with both high accuracy and reliability when given physical parameters of PC nanocavities. It makes rapid and efficient designs in related engineering and applied sciences fields possible and may become a powerful disruptive alternative to existing simulation tools such as FDTD and FEA. The heatmap from transformer blocks also gives some hints for researchers about which parts in their design blueprint will be more important. Our dataset together with code will be released to the community, expecting that it will make a difference for advancing PDA tools in the near future. For this project, we used Pytorch to train the neural networks in the conda environment (anaconda3 2021.11 + Python 3.10.1), where the workstation is equipped with an Intel i7-11800H CPU, an Nvidia GeForce RTX3070 GPU and 16G memory. The information on the manufacturers of these instruments are in the Appendix A. The average time to run one experiment is about 18 to 20 min.

As for limitations of this work, although the best correlation coefficient $V_{coeff}$ is above 90%, chances are that the numbers would be higher if the POViT model was further fine-tuned. For example, hyperparameters including the learning rate, the depth of the transformer encoder, and dimensions of heads in the attention layer can be adjusted for better performance if possible. More importantly, DL models are data-hungry: more data samples with various features will improve prediction accuracies. Furthermore, the data this work used are relatively limited to certain ranges: e.g., simulated *Q* factors are below $5\times 10^{5}$ and modal volumes V>0.8. If more data with larger *Q* and smaller *V* are added into the dataset, POViT will be more robust and generalizable. In the future work, we will enlarge our dataset and conduct more trials with different hyperparameters and algorithms. We hope $V_{coeff}$ can reach above 95% so that POViT can become a reliable and trustworthy simulation tool for researchers in PICs and save them more time than using conventional modeling means. Furthermore, modifications on the proposed POViT model are also expected for improvement in the following stage. Lastly, deep reinforcement learning and transfer learning are currently being explored by us to enable fully autonomous EDA-like optimizing tools for nanolasers and PICs.

## Figures and Tables

**Figure 1 nanomaterials-12-04401-f001:**
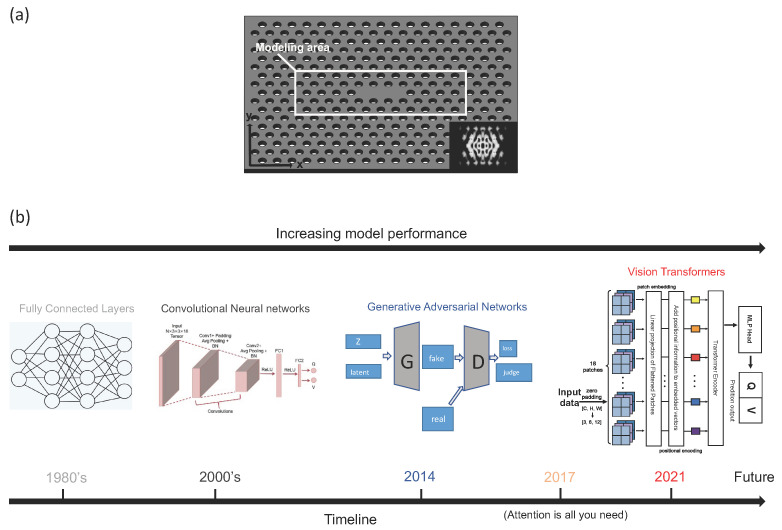
(**a**) structure of the L3 PC nanocavity, constituting a candidate for a typical semiconductor laser. Overall size of the device is around 100 microns in length and 200 nm in thickness. The 54 air holes in the center box will be converted into a 2-D pixel array as the input tensor. Inset at the bottom right: resonant TE mode solved by FDTD simulation. (**b**) evolution of DL models, spanning from the 1980 to the present time. Chronologically, DNNs have gradually evolved from the earliest MLP to CNN, then to Generative Adversarial Networks (GAN), and finally to the latest ViT based on self-attention. ViT has been a disruptive force in CV and image classification etc. since 2021.

**Figure 2 nanomaterials-12-04401-f002:**
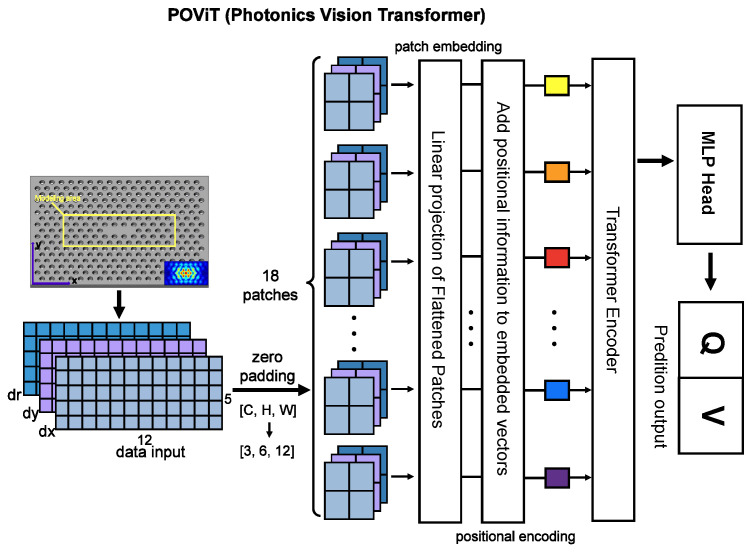
Architecture of the POViT where input tensor shape is N×3×5×12 and output tensor shape is N×2. Input is our PC nanolaser converted into images while output is the predicted *Q* and *V*. Also shown are the transformer encoder with attention, positional encoding as well as patch embedding.

**Figure 3 nanomaterials-12-04401-f003:**
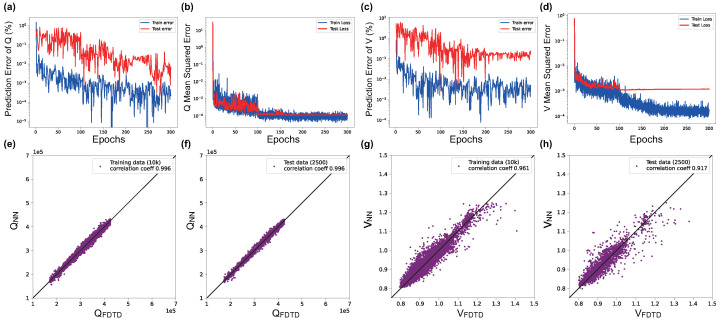
Learning curves and training results for POViT when ABS is used as the activation function. The four plots on the left ((**a**): prediction error, (**b**): MSE, (**e**,**f**): correlation coefficients) are for *Q*, while the four on the right ((**c**): prediction error, (**d**): MSE, (**g**,**h**): correlation coefficients) are for *V*. $Q_{NN}$ is the predicted correlation coefficient, while $Q_{FDTD}$ is the target correlation coefficient. Similarly, $V_{NN}$ and $V_{FDTD}$ are the predicted and target correlation coefficient, respectively.

**Figure 4 nanomaterials-12-04401-f004:**
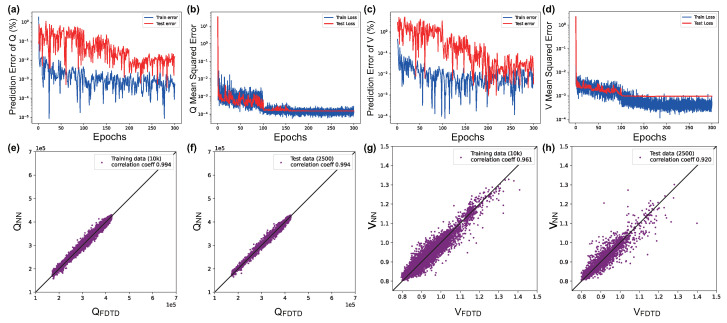
Learning curves and training results for POViT when GELU is used as the activation function. The four plots on the left (**a**,**b**,**e**,**f**) are for *Q*, while the four on the right (**c**,**d**,**g**,**h**) are for *V*.

**Figure 5 nanomaterials-12-04401-f005:**
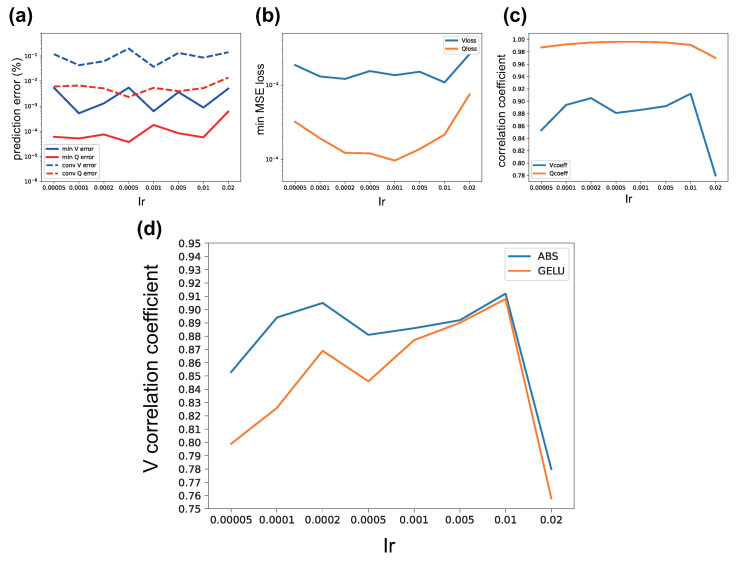
Model performance as a function of learning rates (lr). (**a**): prediction errors vs. lr, conv stands for converged; (**b**): minimum MSE loss vs. lr; (**c**): correlation coefficients vs. lr; (**d**): $V_{coeff}$ calculated with activation functions ABS and GELU vs. lr, respectively. For each plot, eight lr values ranging from 0.00005 to 0.02 are chosen. Data points are mean values between three separate runs.

**Figure 6 nanomaterials-12-04401-f006:**
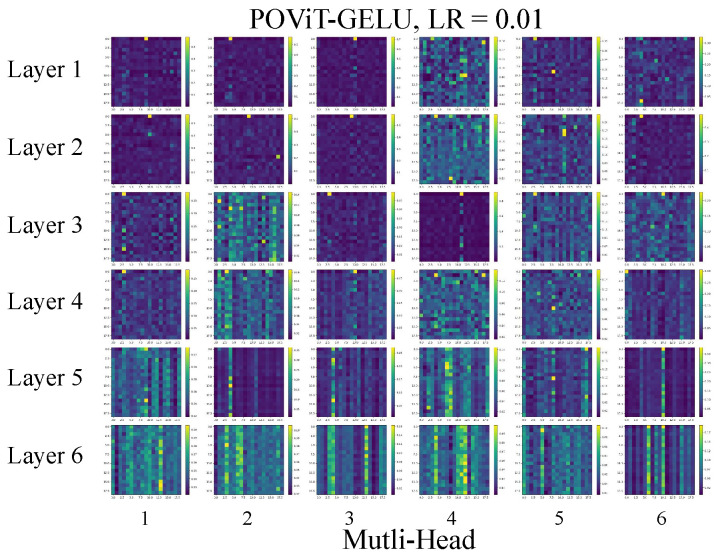
Heatmaps of self-attention in the six-layer POViT model. Learning rate was taken to be 0.01 and GELU was used as the activation function.

**Table 1 nanomaterials-12-04401-t001:** Comparison of performance metrics for *Q* and *V* across several models: CNN, MLP, POViT (ours), and CvT. Conv stands for converged. Coeff values are rescaled to percentages. All the data, except for those cited, are calculated in the process of producing this work. Except for POViT, which records mean values with uncertainties, data for all the other models are best/optimal values. Raw data used for generating this table are displayed in Appendix A in the Appendix A.

Metrics of Q	CNN [8,14]	CNN ^a^	MLP [8]	POViT (Ours)	CvT
MSE Q	0.000247	0.000160	0.000328	0.000116±0.000009	0.000140
Min pred err Q	0.000147%	0.000350%	0.000380%	0.000035±0.000023%	0.000106%
Conv pred err Q	0.018953%	0.011352%	0.004676%	0.002953±0.000618%	0.009752%
Coeff Q	99.0%	98.7%	98.6%	99.5±0.1%	99.4%
**Metrics of V**					
MSE V	0.002520	0.001410	N/A ^b^	0.001114±0.000244	0.001458
Min pred err V	0.003170%	0.003980%	N/A	0.000961±0.000265%	0.002630%
Conv pred err V	0.055800%	0.068351%	N/A	0.036331±0.002708%	0.199203%
Coeff V	N/A	80.5%	N/A	90.8±1.8%	88.8%

^a^ Reproduced experimental results of CNN using code provided in [8]. ^b^ N/A means no data available.

## Data Availability

The data and source code that support the findings of this study are openly available in GitHub at https://github.com/Ping-500ms/POViT.git, (accessed on 20 October 2022).

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
