# Peer review of "POViT: Vision Transformer for Multi-Objective Design and Characterization of Photonic Crystal Nanocavities"

_nanomaterials, 2022, doi:10.3390/nano12244401_

Round 1

Reviewer 1 Report

The important thing of photonic crystal fabrications is considered how to create the nanometer size structures.  The reported computational design and evaluation methods will suggest interesting ideas.  However, in the submitted paper, it was not clear if the obtained results will suggest effective advice for process control or not. As an reviewer, I recommend to that the author should add that some paragraphs to describe abut process streamlining.

Reviewer 2 Report

This paper applied ViT models to Photonic Crystal Nanocavities. The general idea is to solve the problem using the computer vision models. I have several concerns about this paper.

1) There are lots of discussions of deep learning history in the introduction, which makes the readers hard to get the main motivation/story of this paper. 

2) The core idea of the proposed paper is attention, which has been widely used in previous works such as "Dual Attention Matching for Audio-Visual Event Localization, ICCV 19". The authors should discuss the existing attention works in the revision.

3) Missing motivations. Why absolute-value function  is better than GELU?

Round 2

Reviewer 2 Report

The new version addressed my concerns.